# The Prevalence and Clinical Impact of Transition Zone Anastomosis in Hirschsprung Disease: A Systematic Review and Meta-Analysis

**DOI:** 10.3390/children10091475

**Published:** 2023-08-30

**Authors:** Hosnieya Labib, Daniëlle Roorda, J. Patrick van der Voorn, Jaap Oosterlaan, L. W. Ernest van Heurn, Joep P. M. Derikx

**Affiliations:** 1Department of Pediatric Surgery, Amsterdam Gastroenterology and Metabolism Research Institute, Emma Children’s Hospital, Amsterdam UMC, University of Amsterdam and Vrije Universiteit Amsterdam, 1105 AZ Amsterdam, The Netherlands; hosnieya.labib@amsterdamumc.nl (H.L.); d.roorda@amsterdamumc.nl (D.R.); e.vanheurn@amsterdamumc.nl (L.W.E.v.H.); 2Follow Me Program & Emma Neuroscience Group, Department of Pediatrics, Amsterdam Reproduction and Development, Emma Children’s Hospital, Amsterdam UMC, University of Amsterdam, 1105 AZ Amsterdam, The Netherlands; j.oosterlaan@amsterdamumc.nl; 3Department of Pediatric Surgery, Amsterdam Reproduction and Development Research Institute, Emma Children’s Hospital, Amsterdam UMC, University of Amsterdam and Vrije Universiteit Amsterdam, 1105 AZ Amsterdam, The Netherlands; 4Department of Pathology, Amsterdam UMC, Vrije Universiteit Amsterdam, 1081 HV Amsterdam, The Netherlands; jp.vandervoorn@amsterdamumc.nl

**Keywords:** hirschsprung disease, histopathology, constipation, fecal Incontinence, enterocolitis

## Abstract

Background: Hirschsprung disease (HD) is characterized by absent neuronal innervation of the distal colonic bowel wall and is surgically treated by removing the affected bowel segment via pull-through surgery (PT). Incomplete removal of the affected segment is called transition zone anastomosis (TZA). The current systematic review aims to provide a comprehensive overview of the prevalence and clinical impact of TZA. Methods: Pubmed, Embase, Cinahl, and Web of Sciences were searched (last search: October 2020), and studies describing histopathological examination for TZA in patients with HD were included. Data were synthesized into aggregated Event Rates (ER) of TZA using random-effects meta-analysis. The clinical impact was defined in terms of obstructive defecation problems, enterocolitis, soiling, incontinence, and the need for additional surgical procedures. The quality of studies was assessed using the Newcastle–Ottawa Scale. Key Results: This systematic review included 34 studies, representing 2207 patients. After excluding series composed of only patients undergoing redo PT, the prevalence was 9% (ER = 0.09, 95% CI = 0.05–0.14, *p* < 0.001, I^2^ = 86%). TZA occurred more often after operation techniques other than Duhamel (X^2^ = 19.21, *p* = <0.001). Patients with TZA often had obstructive defecation problems (62%), enterocolitis (38%), soiling (28%), and fecal incontinence (24%) in follow-up periods ranging from 6 months to 13 years. Patients with TZA more often had persistent obstructive symptoms (X^2^ = 7.26, *p* = 0.007). Conclusions and Inferences: TZA is associated with obstructive defecation problems and redo PT and is thus necessary to prevent.

## 1. Introduction

Hirschsprung disease (HD) is characterized by absent neuronal innervation of the distal colonic bowel wall for varying distances. The junction between normal ganglionic bowel and aganglionic bowel is characterized by a hypodense and/or non-circumferential distribution of ganglions and hypertrophic nerve fibers called the transition zone (TZ) [1,2] Treatment for HD consists of surgical removal of the aganglionic segment and the transition zone, followed by restoration of bowel continuity with a pull-through (PT) procedure. Different operation techniques can be used to restore bowel continuity, either by creating a pouch (used in Duhamel) or by creating a straight anastomosis (used in all other techniques) [3,4,5,6]. Transition zone anastomosis (TZA) is an incomplete removal of the transition zone and/or aganglionic segment [7]. The occurrence of TZA can be the result of errors in obtaining or interpreting frozen section biopsies. There may also be a discrepancy between the radiological localization of the transition zone, the intra-operative optical transition between dilated and narrowed bowel, and the histopathological localization of the transition zone [8,9,10]. This discrepancy may lead to a false presumption about the localization of the proximal border of the TZ during surgery, resulting in a TZA. A previous systematic review analysing patients who underwent redo surgery for HD reports that abnormal histological findings on repeated biopsies occurred in an estimated 35% of patients who underwent redo pull-through [11]. However, only patients undergoing redo surgery were selected for this systematic review. Selecting only patients undergoing redo surgery therefore may have led to an overestimation of the prevalence of TZA. Furthermore, the clinical symptoms that are associated with a TZA in general remain unclear from this systematic review. Previous studies have suggested that TZA is associated with persistent obstructive defecation problems, soiling, fecal incontinence, and the development of Hirschsprung-associated enterocolitis (HAEC), but it remains unclear what the clinical impact of TZA is, and whether TZA necessitates redo surgery in all patients [12,13,14,15,16].

The primary aim of this systematic review and meta-analysis was to provide a comprehensive overview of the prevalence of TZA. The secondary aim was to provide insight into the clinical impact of TZA after initial corrective surgery in terms of the occurrence of obstructive defecation problems, HAEC, soiling, fecal incontinence, and the need for redo pull-through. This review hypothesizes that a TZA is associated with persistent obstructive defecation problems and requires a redo PT.

## 2. Methods and Materials

The Preferred Reporting Items for Systematic Review and Meta-Analyses (PRISMA) guidelines were used for the design and report of this systematic review [17].

### 2.1. Eligibility Criteria

Studies that met the following criteria were included: (1) describing individuals with a histopathologically confirmed diagnosis of HD, (2) describing assessments of histopathological results (of either biopsy or resection samples) after initial pull-through, or in patients who underwent redo surgery (3) with an observational or case–control design, (4) that were published in a peer-reviewed journal and (5) of which the full-text was available. We excluded studies from this systematic review that were (1) case series with fewer than ten patients, (2) reviews or letters to editors, (3) studies measuring other outcome data, and (4) studies in a language other than English or Dutch.

### 2.2. Search Strategy and Study Selection

The search strategy was designed together with a clinical librarian. Pubmed, Embase, Cinahl, and Web of Sciences were searched, using both simple search terms and hierarchical family forms (e.g., Medical Subject Headings, Thesaurus, Emtree). Three groups of search terms and their equivalents were used: (1) aganglionosis OR Hirschsprung disease, AND (2) pull-through OR anastomosis OR redo-surgery, AND (3) histopathology OR transition zone OR biopsy. The reference lists of eligible articles were also screened for additional articles. The last search was conducted in October 2020. The full search strategy is presented in Appendix A.

Titles and abstracts were screened for eligibility by two authors (HL and DR) using Rayyan, a web-based application for the selection of abstracts [18], followed by a full-text review of the selected articles by the same authors using Covidence [19]. Conflicts in the selection process were solved by consensus, or, when consensus could not be reached, a third party was consulted (JD). Data were extracted by two authors (HL and DR).

### 2.3. Outcomes and Definitions

The primary outcome of this study was the prevalence of TZA. We considered TZA to be present in case of abnormal neuronal innervation of the proximal side of the anastomosis after initial pull-through, including residual aganglionosis and transition zone (characterized by hypoganglionosis and/or abnormal distribution of ganglions). Abnormalities of neuronal innervation had to be confirmed by histopathological examination, either by rectal biopsy (standard or full thickness biopsy) or in the resection specimens. An overview of the definitions and methods of determining a TZA is described in Appendix A (see Appendix A). To compare the prevalence of TZA by operation techniques, we differentiated between pull-through techniques, including Duhamel and all others (such as transanal endorectal pull-through (TERPT), laparoscopic-assisted endorectal pull-through (LEPT), posterior sagittal anorectoplasty (PSARP), pull-through according to Soave, pull-through according to Swenson and pull-through according to Rehbein).

The secondary outcome was to study the clinical impact of TZA after the initial pull-through. Clinical impact was defined as the occurrence of obstructive defecation problems, the occurrence of at least one episode of HAEC, the occurrence of soiling, the occurrence of fecal incontinence, and the need for additional surgical procedures. We adhered to definitions that were used in the included studies to define these outcomes.

### 2.4. Quality Assessment and Risk of Bias in Individual Studies

Two authors (HL and DR) independently assessed the quality of evidence using the Newcastle–Ottawa Scale (NOS) [20]. The NOS allows quantification of the quality of observational studies based on the methods of selecting cases (4 points), comparability of case and control groups (2 points), and outcome measurements (3 points), resulting in a total score ranging from 0 to 9. Following the Agency for Healthcare Research and Quality (AHRQ) standards [21], the quality of studies was considered: good, fair, or poor. Good quality: 3 or 4 stars in selection domain AND 1 or 2 stars in comparability domain AND 2. Fair quality: 2 stars in selection domain AND 1 or 2 stars in comparability domain AND 2 or 3. Poor quality: 0 or 1 star in selection domain OR 0 stars in comparability domain OR 0 or 1 stars. Rating discrepancies were resolved by consensus by including a third party in the discussion.

### 2.5. Statistics

The overall prevalence of TZA was assessed using Comprehensive Meta-Analysis (CMA). As the main summary measure for the prevalence, we employed the aggregated event rate (ER) by aggregating the event rates of TZA in all studies using the random-effects model. The prevalence of TZA was separately aggregated for studies that only included patients undergoing redo surgery and for studies that included patients after initial pull-through. This difference in prevalence was compared using the test of subgroup differences (*Q*) in CMA. We compared the occurrence of TZA after Duhamel with the occurrence of TZA after all other techniques using Chi-square testing. Meta-regression analysis was used to test the association between the publication year of each study and the event rate of TZA.

When the study data allowed for this, the listed clinical symptoms were compared between patients with and without TZA using Chi-square testing. Additionally, we assessed the number of patients that had these symptoms before and after redo pull-through for a TZA using Chi-square testing. Factors were only tested if reported in at least ten studies.

Heterogeneity was interpreted as small (I^2^ ≤ 0.25), medium (I^2^ = 0.25–0.50), or strong (I^2^ ≥ 0.50) according to Higgins [22]. The possibility of publication bias was assessed by visual inspection of Funnel plots and by calculating Funnel plot asymmetry expressed as the Eggers regression intercept *t* [23]. In all statistical analyses, an alpha-level of 0.05 was considered statistically significant.

## 3. Results

### 3.1. Study Population

The search yielded 674 records corresponding to 530 unique articles, of which 34 were included (Figure 1; Flow diagram of the selection process of this systematic review and meta-analysis). Table 1 describes the characteristics of the 34 studies representing 2207 patients, of whom 1516 (69%) were male.

In 19 of the 34 studies, a cohort of patients after initial surgery was described, whilst 15 studies described a cohort of patients who underwent redo pull-through.

The operation techniques used for initial pull-through (n = 31 studies) were Duhamel (n = 712 patients), Soave (n = 513 patients), TERPT (n = 415 patients of whom 52 were one stage and 24 were LEPT), Swenson (n = 138 patients), Rehbein (n = 126 patients), Martin (n = 13 patients), Scott-Bolley (n = 6 patients), missing (n = 208). Three studies did not provide information regarding the used techniques.

### 3.2. Prevalence of TZA

A total of 34 studies were included in the meta-analysis on the prevalence of TZA after the initial pull-through, representing 2207 patients. After the exclusion of the studies that only included patients who underwent redo pull-through, 19 studies were included in the meta-analysis on the prevalence of TZA, representing 1817 patients. The aggregated event rate of the occurrence of TZA was 9% (ER = 0.09, 95% CI = 0.05–0.14, *p* < 0.001, I^2^ = 86%). When including all studies on the prevalence of TZA, the aggregated event rate of the occurrence of TZA was 25% (ER = 0.25, 95% CI = 0.16–0.37, *p* < 0.001, I^2^ = 92%). A forest plot is shown in Figure 2. When including only the 15 studies representing 390 patients who underwent redo pull-through, the aggregated event rate of the occurrence of TZA in these studies was 59% (ER = 0.59, 95% CI = 0.46–0.70, *p* < 0.001, I^2^ = 73%). There was a significant difference in the aggregated event rate of TZA between studies composed of only patients who underwent redo pull-through compared to studies composed of patients after initial pull-through (Q = 49.9, *p* < 0.001). Stratification by operation technique (Duhamel/other techniques) led to the aggregation of 24 studies that allowed for this comparison. This analysis showed that 63 of 641 patients who underwent Duhamel had TZA (10%), whereas 199 of 1137 patients who underwent a different technique than Duhamel had TZA (18%), (*X^2^* = 19.21, *p* < 0.001). The event rate of TZA was not associated with the year of publication of each study (*b* = 0.84, *p* = 0.176).

### 3.3. Clinical Symptoms Associated with a TZA

Table 2 shows the clinical outcomes of patients with and without TZA, as reported in each included study.

Obstructive defecation problems were reported in 62% of patients with TZA after initial surgery (151 of 243 patients, in 19 studies) after a mean follow-up that ranged between six months and 13 years after initial pull-through. Only three studies compared complaints of obstructive defecation in patients with and without TZA after initial surgery and showed that 15 of 56 patients with TZA had constipation (27%) after the initial pull-through and 20 of 170 patients without TZA had constipation after the initial pull-through (12%), indicating that prevalence of obstructive defecation problems is significantly higher in patients with TZA (*X*^2^ = 7.26, *p* = 0.007). Two studies compared complaints of obstructive defecation problems in patients with TZA before and after their redo surgery, and both studies reported complaints of obstructive defecation vanished in all 27 patients after redo surgery.

Postoperative HAEC after initial pull-through was reported in 38% of patients with TZA (67 of 177 patients, in 11 studies) after a mean follow-up that ranged between studies from 6 months to 13 years after initial pull-through. Four studies compared patients with and without TZA concerning HAEC incidence and reported that at least one episode of postoperative enterocolitis occurred in 19 of 75 patients with TZA (25%) and in 33 of 182 patients without TZA (18%), indicating that the risk of HAEC was not significantly higher for patients with TZA (*X*^2^ = 1.71, *p* = 0.191).

Soiling after initial surgery was reported in 28% of patients with TZA (29 of 103 patients in six studies) after a mean follow-up that ranged between studies from 13 months to 10 years after the initial pull-through. No studies compared patients with and without TZA or soiling before and after redo pull-through in patients with TZA.

Fecal incontinence after initial surgery was reported in 24% of patients with TZA (22 of 90 patients in four studies) after a mean follow-up that ranged between studies from 6 to 12 years after the initial pull-through. One study compared patients with and without TZA concerning fecal incontinence after initial pull-through and reported more fecal incontinence in patients with TZA (2 of 30 patients) compared to patients without TZA (0 of 71 patients), but did not statistically test this difference.

Of the 389 patients with TZA in this systematic review, 323 underwent redo pull-through (83%). All indications for redo pull-through as described in the included studies are reported in Appendix A (see Appendix A). Some patients had multiple indications for one procedure of redo surgery. Additional surgical interventions after the initial pull-through, other than redo pull-through, were indicated in 42% of patients with TZA (74 of 178 patients in 11 studies).

### 3.4. Quality of Evidence and Risk of Bias Analyses

The quality of studies, as assessed with the NOS, is reported in Table 3. Among the 34 articles, 26 articles were of poor quality, 3 were of fair quality, and only 3 were of good quality. The average quality score was 6.5 (range of 6–8). The most common reason for poor quality was the use of an uncontrolled design: these studies carried the risk of selection bias and allocation bias, as these studies only described a series of patients after redo surgery. The risk of observer bias was also present, as none of the studies were blinded. There was no risk of publication bias based on visual inspection of the funnel plot (see Appendix A, in Appendix A) and based on Egger’s regression (*t =* 0.486, *p* = 0.711).

## 4. Discussion

This systematic review and meta-analysis showed that the overall prevalence of TZA was 25%, but varied between series describing only patients who underwent redo pull-through (59%) and series describing its occurrence after initial pull-through (9%). A large part of the studies in the current systematic review consisted of only patients undergoing redo pull-through, which may have led to an overestimation of the prevalence of TZA in the general population, as TZA is often an indication of redo PT. We thus think the actual prevalence may be best reflected by the prevalence of 9%.

Our prevalence findings were lower than in a previous meta-analysis conducted by Friedmacher et al., who reported TZA in 35% of patients [11]. This difference may be explained by differences in case definition, differences in inclusion criteria, and the inclusion of studies (13 studies) that were published after their study period. Our sensitivity analysis in studies with patients who underwent redo PT showed an even higher prevalence, which may suggest that over the past decade, there has been an increased awareness of the risk of TZA and thus a higher detection rate of TZA [57]. However, our data did not show a significant increase in prevalence over the years of our study period. The most accurate way to estimate the actual prevalence of a TZA in all patients who receive pull-through surgery would be based on the histopathological report of the proximal resection plane, but only a few of the included studies in the current systematic review describe this for all patients [58].

Several factors account for challenges in the diagnosis of a TZA. An important factor is the current approach to determining the delineation of the transition zone during operation [8]. Most surgeons achieved this by perioperative single-point frozen section, based on radiological and visual assessment of the location of the tapering of the bowel. However, previous studies show the limited validity of single-point biopsy and that the radiological and perioperative assessment of a surgeon does not correspond well to the histopathological delineation of the transition zone of the resected bowel [59,60]. Another important factor is the lack of a clear definition of transition zone bowel. Although insight into histopathological features of the transition zone is increasing [2,61,62], with an increasing number of studies describing histopathological findings in transition zone bowel, a clear definition of the histopathological criteria of the transition zone is still lacking [57,63,64]. Moreover, studies that describe histopathological features of the transition zone have shown large variation between patients concerning the length of the transition zone, the skewness of its proximal delineation, and the distribution of ganglions and nerve fibers in the transition zone [2,61,65,66]. These features are more likely diagnosed after calretinin immunohistochemistry, which requires time and therefore cannot be performed in frozen perioperative biopsy specimens [67]. In current practice, we often also determine the presence of ganglion cells in the proximal resection plane specimen. Despite this, in a substantial number of patients in the studies included in this review, TZA was detected later in life by repeat biopsy. This emphasizes that not only the presence of ganglion cells may be an important feature to take into account, but also the density and distribution of ganglion cells and hypertrophy of nerve fibers [2,61].

Our findings suggest that a TZA occurs more often when techniques other than Duhamel are used. It is the purpose of the pouch (which is also aganglionic) to slow down the passage of stools to become more solid before defecation, and thus not lead to constipation or incontinence. This, in turn, may explain the lower rates of diagnosis of TZA in patients with a pouch.

The second aim of this review was to study the clinical impact of a TZA. The evidence presented in this systematic review shows the more frequent occurrence of obstructive defecation problems, enterocolitis, soiling, incontinence, and redo surgery in patients with TZA compared to patients without TZA. A direct comparison between patients with and without TZA showed more obstructive defecation problems in patients with TZA compared to patients without TZA, but could only be based on three studies [27,30,47] Thus, conclusions need to be drawn with caution. However, the rates of obstructive defecation problems, HAEC, soiling, and fecal incontinence in patients with TZA in the current systematic review seem to be higher compared to what has been described in previous meta-analyses on outcomes after pull-through surgery in patients with HD in general [13,14]. Note that this comparison has to be drawn with caution, given the inclusion of 15 (44%) series describing redo procedures and the limited number of studies in this analysis comparing patients before and after redo PT for TZA. Furthermore, soiling can be caused by TZA, resulting in a low obstruction and thus overflow diarrhea. However, it can also be the result of damage to the sphincter complex. Therefore, it is hard to conclude what the actual cause of soiling is in the included studies. Moreover, poor bowel function can be the result of obstructive complaints and incontinence (overflow) or a combination of the two. Constipation and incontinence can both occur in patients and can be the result of problems in functional defecation. We considered separately and adhered to the definition used in the included studies. Meta-analysis limits us in further exploring, on a patient level, what patient symptoms co-occurred.

Lastly, the findings of the current systematic review suggested that patients with a TZA often received redo pull-through (83%) or other types of surgical interventions. We noticed that about half of the redo procedures described in this review were performed because of a TZA.

To conclude, our findings suggest that TZA accounts for substantial morbidity in patients with HD, and, when diagnosed, it often results in redo pull-through. To improve outcomes and aid in predicting the need for redo pull-through procedures, it is important to detect TZA early.

### 4.1. Limitations

The interpretations of our findings on the prevalence and clinical impact of TZA need to be considered in light of the following limitations. First, the aforementioned challenge is diagnosing TZA. Second, we used a liberal definition for TZA in this study, including both abnormal histological findings of residual aganglionosis and characteristics of TZ, whilst a clear definition of the histopathological features of the TZ is still lacking. Previous studies have suggested that residual aganglionosis may also be acquired, and may be the result of ischemia instead of the result of an incomplete resection [68]. Therefore, we need a better definition of TZA and more studies should apply a homogeneous definition of TZA. Third, heterogeneity in the current meta-analyses was strong for all meta-analytical findings. There was also large heterogeneity in outcome definitions of clinical symptoms between studies, and large heterogeneity in the length of follow-up. Although this is inherent to a systematic review, it made it challenging to compare clinical outcomes across studies. In particular, concerning fecal incontinence, the definition depends on the age of the patient and whether toilet training has been completed. Lastly, although an effort was made to not include overlapping cohorts of patients, (partly) overlapping patients between studies could not be fully ruled out on a patient level.

### 4.2. Quality of Evidence and Risk of Publication Bias

The quality of evidence retrieved from the majority of studies (65%) was poor, as was reflected by NOS scores. Most studies had a retrospective design and therefore a lot of relevant data on outcome and per-operative details could not be retrieved from the studies. Due to the poor quality of the studies, the findings are difficult to generalize. There was, however, no significant risk of publication bias.

### 4.3. Clinical Implications and Future Perspectives

These aforementioned limitations highlight the need for high-quality studies reporting the prevalence and clinical impact of a TZA in patients with HD. For future studies, we recommend studies that focus on the histopathological features of (the proximal delineation of) the TZ, allowing for a clear definition of the occurrence of TZA, a better approach to determining the perioperative level of resection, and, most importantly, prevention of TZA at the time of the initial procedure. We further recommend that studies report bowel function using more uniform definitions and not focus on single outcomes only, but report outcomes following a recently developed core outcome set for patients with HD [69]. This will allow for a better comparison of clinical impact in patients with a TZA compared to patients without a TZA and a better comparison and aggregation of outcomes between studies. Ultimately, this will help us answer the question of whether redo pull-through is necessary in all patients with a TZA.

## 5. Conclusions

In this study, we found that the prevalence of TZA was 25% in patients with HD after the initial pull-through. Prevalence was 9% when excluding series that included only patients undergoing redo PT (of which most were conducted because of TZA). Patients with TZA had a high prevalence of obstructive defecation problems (61%), enterocolitis (38%), soiling (28%), fecal incontinence (24%), and redo pull-through (83%).

## Figures and Tables

**Figure 1 children-10-01475-f001:**
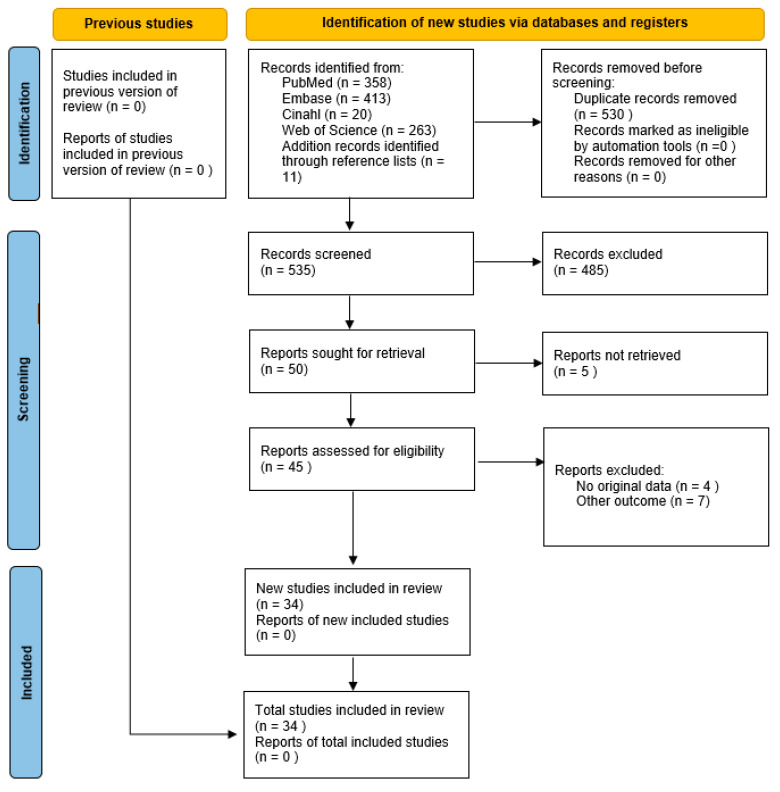
Flow diagram of the selection process of this systematic review and meta-analysis.

**Figure 2 children-10-01475-f002:**
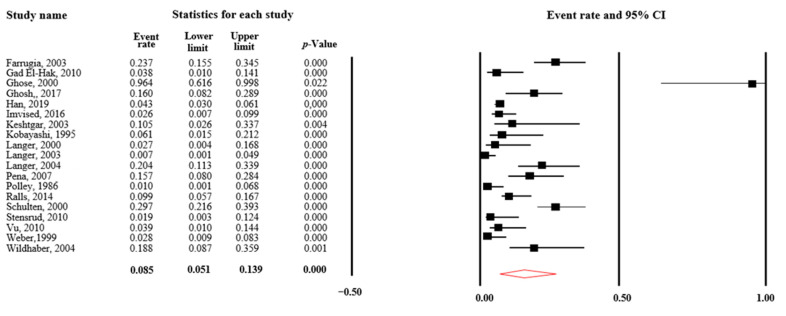
Forest plot of the event rates (ER) of transition zone anastomosis in patients with Hirschsprung disease (excluding redo-cohorts) [16,27,28,29,30,33,35,36,37,38,39,41,45,46,47,50,52,53,55].

**Table 1 children-10-01475-t001:** Characteristics of included studies.

Study	Sample Size, n	Male (%)	Mean Age (SD) Initial Corrective Surgery in Months	Mean Age (SD) Redo Surgery in Months	Mean (SD) Follow-Up in Months after Redo	Length of Aganglionic Segment	Operation Techniques Initial Corrective Surgery	Number of Patients with Redo Surgery, n (%)	Operation Techniques Redo Surgery	Number of TZA, n (%)	Number of Patients with TZA of the Patients with Redo Surgery, n (%)
Chatoorgoon, 2011 [24]	17	NR	NR	84	24	NR	17 Duhamel	17/17 (100%)	8 Posterior sagittal approach, 9 transanal Swenson	8/17 (47%)	8/17 (47%)
Coe, 2012 [25]	30	73%	10.2 (8.21)	67.0 (42.8)	24.8 [18]	9 short, 3 long, 6 total colonic, 7 NR	17 Soave, 11 Duhamel, 2 Swenson	30/30 (100%)	NR	19/30 (63%)	19/30 (63%
Dingemans, 2017 [26]	16	63%	NR	NR	NR	NR	8 Soave, 2 Swenson, 3 Rehbein, 3 Duhamel	16/16 (100%)	15 Swenson, 1 Soave	8/16 (50%)	8/16 (50%)
Farrugia, 2003 [27]	76	80%	NR	NR	6.0 (4.1)	56 short, 15 long, 5 total colonic	76 Duhamel	5/76 (7%)	NR	18/76 (24%)	3/5 (60%)
Gad El-Hak, 2010 [28]	52	63%	3.29 (1.6)	NR	NR	NR	52 Swenson	2/52 (4%)	2 Swenson	2/52 (4%)	2/2 (100%)
Ghose, 2000 [29]	13	77%	NR	NR	118	10 short, 3 long	8 Duhamel, 5 Soave	7/13 (54%)	7 Duhamel	13/13 (100%)	7/7 (100%)
Ghosh, 2017 [30]	50	78%	NR	NR	NR	36 short, 14 long	50 Soave	0/50 (0%)	NR	8/50 (16%)	NR
Gupta, 2019 [31]	32	81%	NR	62.4	43.2	28 short, 24 long, 2 total colonic	24 Duhamel, 6 Scott Boley, 2 Swenson	32/32 (100%)	22 open TEPT Scott Boley/Soave, 5 ileoanal PT, 5 TEPT	12/32 (38%)	12/32 (38%)
Hadidi, 2007 [32]	18	78%	NR	NR	NR	NR	7 Swenson, 9 Soave, 1 Duhamel, 1 Rehbein	18/18 (100%)	18 TEPT	18/18 (100%)	18/18 (100%)
Han, 2019 [16]	657	76%	NR	NR	59.6 (53.76)	580 short, 77 long	443 Duhamel, 199 TEPT (Soave like), 12 Martin, 3 Swenson	57/657 (9%)	29 Duhamel, 23 TEPT, 4 Swenson, 1 PSAR	28/657 (4%)	28/57 (49%), 8/28 (29%) underwent 3 times
Imvised, 2016 [33]	76	87%	9.9	NR	NR	73 short, 3 long	76 TEPT	0/76 (0%)	NR	2/76 (3%)	0/76 (0%)
Jiang, 2019 [34]	31	71%	5	41	NR	NR	18 TEPT, 5 Duhamel, 8 Soave	31/31 (100%)	14 Duhamel, 17 Soave	16/31 (52%)	16/31 (52%)
Keshtgar, 2003 [35]	19	79%	NR	NR	NR	NR	16 Duhamel, 1 Rehbein, 2 Soave	5/19 (26%)	NR	2/19 (11%)	1/5 (20%)
Kobayashi, 1995 [36]	33	85%	NR	NR	29	27 short, 6 long	31 Swenson, 2 Duhamel	1/33 (3%)	NR	2/33 (6%)	1/1 (100%)
Langer, 2000 [37]	37	73%	1.0 & 2.0 & 2.5	NR	NR	31 short, 6 long	13 open Soave, 24 transanal Soave	7/37 (19%)	NR	1/37 (3%)	1/7 (14%)
Langer, 2003 [38]	141	80%	4.9	NR	20.2 (9.2)	109 short, 32 long	141 1-stage transanal Soave	3/141 (2%)	1 Duhamel, 2 Soave	1/141 (1%)	1/3 (33%)
Langer, 2004 [39]	49	84%	NR	NR	NR	27 short, 22 long	9 Swenson, 25 soave, 15 Duhamel	17/49 (35%)	16 Duhamel, 1 Swenson	10/49 (20%)	8/17 (47%)
Lawal, 2011 [40]	16	88%	NR	NR	NR	11 short, 2 long, 3 NR	7 transabdominal Soave, 7 transanal Soave, 1 transanal Swenson, 1 Duhamel	16/16 (100%)	15 TEPT, 1 posterior sagittal approach	16/16 (100%)	16/16 (100%)
Peña, 2007 [41]	51	67%	68.4	NR	NR	NR	17 Soave, 14 Duhamel, 6 TEPT, 5 Swenson, 1 Swenson J pouch, 1 Soave and right colonic pouch, 7 NR	45/51 (88%)	40 posterior sagittal approach (with or without laparotomy), 5 transanal approach	8/51 (16%)	8/45 (18%)
Peng, 2021 [42]	46	83%	18.34 (30.9)	53.6 (37.2)	101.4 (33.2)	NR	38 Soave, 2 Rehbein, 1 Martin, 5 NR	46/46 (100%)	46 Soave	27/46 (59%)	27/46 (59%)
Pini-Prato, 2010 [43]	70	60%	22.8 (32.4)	48 (37.2)	94.8 (49.2)	21 short, 4 long, 10 total colonic	37 Soave, 7 Duhamel, 1 Swenson, 2 Rehbein, 23 NR	70/70 (100%)	53 Soave, 9 Duhamel, 7 Swenson, 1 Rehbein/posterior sagittal approach	51/70 (73%)	51/70 (73%)
Pini-Prato, 2020 [44]	16	75%	NR	56.7 (37.6)	26	12 short, 2 long, 2 total colonic	NR	16/16 (100%)	NR	10/16 (63%)	10/16 (63%)
Polley, 1986 [45]	99	70%	NR	NR	NR	73 short, 19 long, 6 total colonic	78 TEPT, 11 Swenson, 5 Duhamel, 1 colectomy, 4 NR	12/99 (12%)	4 Swenson, 1 Duhamel, 2 TEPT, 5 NR	1/99 (1%)	1/12 (8%)
Ralls, 2014 [46]	121	74%	26.4 (50.4)	82.8 (110.4)	NR	90 short, 24 long, 7 NR	NR	32/121 (26%)	9 Swenson, 5 Duhamel, 7 transanal, 6 TEPT, 1 Soave, 1 open, 3 NR	12/121 (10%)	12/32 (38%)
Schulten, 2000 [47]	101	73%	22.8	NR	54	NR	101 Rehbein	0/101 (0%)	NR	30/101 (30%)	0/101 (0%)
Schweizer, 2007 [48]	17	NR	NR	NR	126	NR	13 Rehbein, 3 Soave, 1 Duhamel	17/17 (100%)	17 Duhamel	16/17 (94%)	16/17 (94%)
Sheng, 2012 [49]	24	63%	8	NR	30	5 short, 19 NR	9 Duhamel, 12 Soave, 2 Rehbein, 1 Swenson	24/24 (100%)	6 Duhamel, 7 Soave, 3 Rehbein, 1 laparotomy, 7 posterior sagitally	5/24 (21%)	5/24 (21%)
Stensrud, 2010 [50]	52	NR	NR	NR	NR	43 short, 9 long	28 TEPT, 24 LEPT	1/52 (2%)	1 LEPT	1/52 (2%)	1/1 (100%)
Van Leeuwen, 2000 [51]	19	68%	27.6	64.8	165.6	11 short, 5 long, 3 NR	10 TEPT, 5 Duhamel, 3 Swenson, 1 Rehbein	19/19 (100%)	3 Swenson, 8 TEPT, 8 Duhamel	5/19 (26%)	5/19 (26%)
Vu, 2010 [52]	51	65%	0.7 (0.2)	NR	18 (2.4)	47 short, 4 long	51 TOSEPT	2/51 (4%)	NR	2/51 (4%)	2/2 (100%)
Weber, 1999 [53]	107	NR	9	NR	102	93 short, 7 long, 7 total colonic	68 Soave, 39 Duhamel	5/107 (5%)	2 Soave, 2 Duhamel, 1 NR	3/107 (3%)	3/5 (60%)
Wilcox, 1998 [54]	20	80%	10.8	72	78	NR	10 Duhamel, 3 Soave, 7 Swenson	20/20 (100%)	1 Swenson, 19 Duhamel	10/20 (50%)	10/20 (50%)
Wildhaber, 2004 [55]	32	69%	NR	NR	103.2	NR	NR	9/32 (28%)	NR	6/32 (19%)	5/9 (56%)
Xia, 2016 [56]	18	89%	5	38	NR	18 short	18 Soave	18/18 (100%)	15 Soave, 3 Swenson	18/18 (100%)	18/18 (100%)

d = days; m = months; y = year; NR = not reported; TZA = transition zone anastomosis; TEPT = transanal endorectal pull-through; LEPT = laparoscopic endorectal pull-through; PSARP = posterior sagittal anorectoplasty; TOSEPT = transanal one-stage endorectal pull-through.

**Table 2 children-10-01475-t002:** Clinical symptoms described in patients with TZA after initial surgery and after redo surgery.

Study	Number of Patients with TZA	Mean Follow-up after Initial PT	Mean Follow-up after Redo PT	Constipation		Soiling		Enterocolitis		Fecal Incontinence		Continence	
				After Initial Surgery	After Redo Surgery	After Initial Surgery	After Redo Surgery	After Initial Surgery	After Redo Surgery	After Initial Surgery	After Redo Surgery	After Initial Surgery	After Redo Surgery
Chatoorgoon, 2011 [24]	8	7 y	2 m	8/8 (100%)	0/8 (0%)	-	-	-	-	-	-	-	-
Coe, 2012 [25]	19	7.8 y	25 m	19/19 (100%)	0/19 (0%)	-	-	10/19 (53%)	-	-	-	-	-
Dingemans, 2017 [26]	8	7.6 y	3 y	0/8 (0%)	0/8 (0%)	1/8 (13%)	-	-	-	3/8 (38%)	-	-	-
Farrugia, 2003 [27]	18	6 y		6/18 (33%)	-	2/18 (11%)	-	11/18 (61%)	-	4/18 (22%)	-	-	-
Gad El-Hak, 2010 [28]	2	12 m		2/2 (100%)	0/2 (100%)	-	-	-	-	-	-	-	-
Ghose, 2000 [29]	13	10 y		7/13 (54%)	-	5/13 (38%)	2/7 (29%)	1/13 (8%)	-	13/13 (100%)	1/7 (14%)	-	2/7 (29%)
Ghosh, 2017 [30]	8	6 m		2/8 (25%)	-	-	-	4/8 (50%)	-	-	-	-	-
Hadidi, 2007 [32]	18	43 m		-	3/18 (17%)	-	1/18 (6%)	8/18 (44%)	2/18 (11%)	-	-	-	-
Keshtgar, 2003 [35]	2			2/2 (100%)	-	-	-	-	-	-	-	-	-
Kobayashi, 1995 [36]	2	29 m		1/2 (50%)	-	-	-	1/2 (50%)	-	-	-	-	-
Langer, 2004 [39]	10			5/10 (50%)	-	-	-	5/10 (50%)	-	-	-	-	-
Lawal, 2011 [40]	16	6–66 m	16 m	12/16 (75%)	1/16 (6%)	-	1/16 (6%)	9/16 (56%)	0/16 (0%)	-	-	-	-
Pini-Prato, 2010 [43]	51	11.9 y	7.9 y	25/51 (49%)	-	-	-	8/51 (16%)	-	2/51 (4%)	-	-	-
Pini-Prato, 2020 [44]	10		26 m	10/10 (100%)	-	-	-	-	-	-	-	-	-
Polley, 1986 [45]	1	3 m–11 y		-	-	-	-	-	-	-	-	-	1/1 (100%)
Schulten, 2000 [47]	30	4.5 y		6/30 (20%)	-	2/30 (7%)	-	-	-	-	-	-	-
Schweizer, 2007 [48]	16		9 y	16/16 (100%)	3/16 (19%)	13/16 (81%)	2/16 (13%)	9/16 (56%)	-	-	-	-	-
Sheng, 2012 [49]	5	4.5 y	2.5 y	5/5 (100%)	0/5 (0%)	-	2/5 (40%)	-	0/5 (0%)	-	0/5 (0%)	-	-
Stensrud, 2010 [50]	1	5.7 m TEPT, 10 m LEPT		1/1 (100%)	-	-	-	-	-	-	1/1 (100%)	-	-
Wilcox, 1998 [54]	9	12.5 y	6.5 y	-	0/9 (0%)	-	-	-	-	-	-	-	-
Wildhaber, 2004 [55]	6	13.1 y	8.6 y	6/6 (100%)	-	-	-	1/6 (17%)	-	-	-	-	-
Xia, 2016 [56]	18	13–75 m		18/18 (100%)	0/18 (0%)	6/18 (33%)	-	-	6/18 (33%)	-	-	-	18/18 (100%)
Total	261			151/243 (62%)	7/119 (9%)	29/103 (28%)	8/62 (13%)	67/177 (38%)	8/57 (14%)	22/90 (24%)	2/13 (15%)	-	21/26 (81%)

**Table 3 children-10-01475-t003:** Quality assessment of included studies with the Newcastle–Ottawa Scale (NOS).

Study	Selection	Comparability	Outcome	Total Score	Quality
Chatoorgoon, 2011 [24]	3	0	3	6	Poor
Coe, 2012 [25]	3	0	3	6	Poor
Dingemans, 2017 [26]	3	0	3	6	Poor
Farrugia, 2003 [27]	4	1	3	8	Fair
Gad El-Hak, 2010 [28]	3	0	3	6	Poor
Ghose, 2000 [29]	3	0	3	6	Poor
Ghosh, 2017 [30]	4	1	3	8	Fair
Gupta, 2019 [31]	3	0	3	6	Poor
Hadidi, 2007 [32]	3	0	3	6	Poor
Han, 2019 [16]	4	1	3	8	Fair
Imvised, 2016 [33]	3	0	3	6	Poor
Jiang, 2019 [34]	4	0	3	7	Poor
Keshtgar, 2003 [35]	3	0	3	6	Poor
Kobayashi, 1995 [36]	4	0	3	7	Poor
Langer, 2000 [37]	3	2	3	8	Good
Langer, 2003 [38]	3	0	3	6	Poor
Langer, 2004 [39]	4	0	3	7	Poor
Lawal, 2011 [40]	3	0	3	6	Poor
Peña, 2007 [41]	3	0	3	6	Poor
Peng, 2021 [42]	3	0	3	6	Poor
Pini-Prato, 2010 [43]	4	0	3	7	Poor
Pini-Prato, 2020 [44]	3	0	3	6	Poor
Polley, 1986 [45]	3	0	3	6	Poor
Ralls, 2014 [46]	3	2	2	7	Good
Schulten, 2000 [47]	4	0	3	7	Poor
Schweizer, 2007 [48]	3	0	3	6	Poor
Sheng, 2012 [49]	3	0	3	6	Poor
Stensrud, 2010 [50]	3	0	3	6	Poor
Van Leeuwen, 2000 [51]	3	0	3	6	Poor
Vu, 2010 [52]	3	0	3	6	Poor
Weber, 1999 [53]	4	0	3	6	Poor
Wilcox, 1998 [54]	3	0	3	6	Poor
Wildhaber, 2004 [55]	3	0	3	6	Poor
Xia, 2016 [56]	4	1	3	8	Good

## Data Availability

Not applicable.

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
