# Peer review of "The Prevalence and Clinical Impact of Transition Zone Anastomosis in Hirschsprung Disease: A Systematic Review and Meta-Analysis"

_children, 2023, doi:10.3390/children10091475_

Round 1

Reviewer 1 Report

At first, I would like to congratulate the authors for their efforts.

The authors have conducted a systematic review on the prevalence and clinical impact of transition zone anastomosis (TZA) in HD. The authors have included all papers where TZA is present. TZA was defined as abnormal neuronal innervation in the proximal portion of the anastomosis. The authors included both aganglionosis and transition zone in the operational definition of abnormal neuronal innervation. They identified 34 studies, majority were of poor methodological quality. The overall prevalence of TZA was 25%. The prevalence of TZA in redo and fresh pull throughs were 9% and 59% respectively. Patients with TZA had significantly more obstructive symptoms as compared to those without TZA. Other outcomes, incontinence, soiling, HAEC, etc. showed no significant differences among the two groups. . TZA 36 occurred more often after operation techniques other than Duhamel (X2= 19.21, p=<0.001).

The study has merit and will be of interest to our readers. However, I have a few concerns:

Introduction: Please write your hypothesis in 2-3 lines at the end of the Introduction section. What did you hypothesize before conducting this research?

Methods: One of the major issues in this review is the operational definition of TZA. Here, they have included "residual aganglionosis" in the definition of TZA. Residual aganglionosis can be because of many reasons, e.g., poor vascular supply. It doesn't mean that aganglionosis is synonymous with TZ. I am very skeptical about this definition and I am sure the prevalence of 29% is because of this liberal definition. The authors must change the title of paper if they want to stick with this methodology:

The prevalence and clinical impact of residual aganglionosis and transition zone anastomosis in Hirschsprung disease: A systematic review and meta- analysis

Results: Please revise Figure 1 as per the revised PRISMA 2020 guidelines.

In the subheading "Prevalence of TZA", there seems to be a typographical error. Lines no 168-170, both highlight that aggregated prevalence is 9% and 25%. Please correct this. 

Otherwise, the paper is well-written and will be of interest to our readers. The authors must incorporate the above suggestions.

Moderate editing is needed.

Author Response

At first, I would like to congratulate the authors for their efforts.

The authors have conducted a systematic review on the prevalence and clinical impact of transition zone anastomosis (TZA) in HD. The authors have included all papers where TZA is present. TZA was defined as abnormal neuronal innervation in the proximal portion of the anastomosis. The authors included both aganglionosis and transition zone in the operational definition of abnormal neuronal innervation. They identified 34 studies, majority were of poor methodological quality. The overall prevalence of TZA was 25%. The prevalence of TZA in redo and fresh pull throughs were 9% and 25% respectively. Patients with TZA had significantly more obstructive symptoms as compared to those without TZA. Other outcomes, incontinence, soiling, HAEC, etc. showed no significant differences among the two groups. . TZA 36 occurred more often after operation techniques other than Duhamel (X2= 19.21, p=<0.001).

The study has merit and will be of interest to our readers. However, I have a few concerns:

Introduction: Please write your hypothesis in 2-3 lines at the end of the Introduction section. What did you hypothesize before conducting this research?

We would like to thank the reviewer for the feedback and suggestions.

We have now added a hypothesis at the end of our introduction section: ‘This review hypothesizes that TZA is associated with persistent obstructive defecation problems and requires a redo PT.’

Methods: One of the major issues in this review is the operational definition of TZA. Here, they have included "residual aganglionosis" in the definition of TZA. Residual aganglionosis can be because of many reasons, e.g., poor vascular supply. It doesn't mean that aganglionosis is synonymous with TZ. I am very skeptical about this definition and I am sure the prevalence of 29% is because of this liberal definition. The authors must change the title of paper if they want to stick with this methodology:

The prevalence and clinical impact of residual aganglionosis and transition zone anastomosis in Hirschsprung disease: A systematic review and meta- analysis.

We agree with the reviewer about his concerns. We have now changed our title to ‘The prevalence and clinical impact of residual aganglionosis and transition zone anastomosis in Hirschsprung disease: A systematic review and meta- analysis.’

We have decided to use a liberal definition of TZA (and to use broad search terms). The aim of this decision was to provide  as much information as possible for the meta-analysis. We acknowledge this limitation and the need for a better and homogeneous definition We address this now in the limitations section of the discussion.

Please see lines 329-331 of our Discussion section: “Second, we used a liberal definition for TZA in this study, including both residual aganglionosis, as well as TZ. A clear definition of the histopathological features of the TZ is still lacking. Previous studies have suggested that residual aganglionosis may also be acquired, and may be the result of ischemia, instead of the result of an incomplete resection[1]. Uniform  definitions are required to better compare future studies

Results: Please revise Figure 1 as per the revised PRISMA 2020 guidelines.

We have now revised figure 1 as per the revised PRISMA 2020 guidelines.

In the subheading "Prevalence of TZA", there seems to be a typographical error. Lines no 168-170, both highlight that aggregated prevalence is 9% and 25%. Please correct this. 

We have corrected this. Please see lines 305-309:‘After exclusion of the studies that only included patients who underwent redo pull-through, 19 studies were included in the meta-analysis on the prevalence of TZA, representing 1817 patients. The aggregated event rate of the occurrence of a TZA was 9% (ER=0.09, 95%CI=0.05-0.14, p<0.001, I2=86%). When including all studies on the prevalence of TZA, the aggregated event rate of the occurrence of a TZA was 25% (ER=0.25, 95%CI=0.16 -0.37, p<0.001, I2=92%).’

Otherwise, the paper is well-written and will be of interest to our readers. The authors must incorporate the above suggestions.

Reviewer 2 Report

This review manuscript authored by Hosnieya Labib .et.al aim to provide a comprehensive overview of the prevalence and clinical impact of transition zone anastomosis (TZA) in patients with Hirschsprung disease (HD) who undergo pull-through surgery (PT). The abstract is well-written and effectively conveys the key findings and implications of the study. However, the manuscript requires additional revision to adequately address the issues raised in the review. Further revision of interpretation of results, clarity and accuracy of data analysis and figures could greatly enhance the manuscript impact and make it more engaging for readers.

·       ·       The valuable systematic review on transition zone anastomosis (TZA) in Hirschsprung disease (HD) provided insightful findings based on studies available until October 2020. However, considering that there might be more recent research articles and case studies on this topic after 2020, I would like to kindly suggest that the authors consider incorporating the latest results into their manuscript. This addition could further enhance the review's comprehensiveness and relevance to the current scientific literature.

·       Line 75: Please remove “2. Materials and Methods”

·       In line 108, kindly consider adding clarification within parentheses near the term “rectal biopsy.” Specifically, including “(standard and full-thickness rectal biopsy)” would enhance the clarity of the methods being discussed and provide readers with a better understanding of the types of biopsies considered in the study.

·       In the discussion of histopathological procedures for assessing nerve cell abnormalities, I would like to inquire whether the authors have considered the use of immunohistochemistry with antibodies targeting specific markers like S100, Hu, or calretinin. This approach can be valuable in identifying ganglion cells and aiding in the diagnosis of conditions such as Hirschsprung's disease. Including this possibility in the manuscript could enhance the comprehensiveness of the discussion.

·       I kindly recommend that the authors consider relocating all supporting/supplementary tables (sTable1 and sTable2) from the main text to a supplementary data  section. The presence of multiple supporting information tables within the main text might potentially confuse readers. By placing these tables in a dedicated section, the manuscript's organization and readability can be improved, allowing readers to access supplementary information more efficiently.

·       In addition, I would like to suggest including reference numbers in the "authors" column for all the tables. This addition will greatly assist readers in easily identifying the corresponding studies and enhance the overall clarity of the manuscript.

·       In page 5, under the section "Quality assessment and risk bias in individual studies," I would like to kindly point out that while the text mentions the Newcastle-Ottawa Scale (NOS) criteria and their respective score ranges (0 to 4, 2, and 3 points), it does not explicitly state the total possible score range for the NOS. Including this information, which ranges from 0 to 9, will provide readers with a clearer understanding of how the total score reflects the methodological quality of the observational studies.

·       In page 5, under the section "Quality assessment and risk bias in individual studies," I would like to kindly bring to the authors' attention that the text mentions the categorization of study quality as "good, fair, or poor" based on the NOS scores. However, specific cutoff values for defining what constitutes "good," "fair," or "poor" quality are not provided. It would be helpful to include this information to clarify the criteria used for categorizing the studies and to enhance the transparency of the quality assessment process.

·       Furthermore, in the same section, it is mentioned that rating discrepancies were resolved by consensus, which is a standard approach. However, it would be beneficial to provide additional details on how the consensus was reached, such as through discussion, arbitration by a third party, or any other established process. This information would enhance the transparency of the quality assessment process and provide readers with a better understanding of the rigor of the study's methodology.

·       I would like to kindly suggest placing the figure number and legends just below each figure in the manuscript. This arrangement will improve the clarity and readability of the figures, making it easier for readers to understand the content and context of each figure.

·       Request the authors to include figures with their corresponding legends in the manuscript. Additionally, please ensure that each legend provides a clear and comprehensive explanation of the content depicted in the figure. This addition will enhance the reader's understanding and interpretation of the figures, contributing to the overall clarity and quality of the manuscript.

·       I would like to kindly bring to the authors' attention that in the statistical analysis section, it is mentioned that publication bias was assessed using funnel plot asymmetry and quantified with Egger's intercept. However, the data regarding the Egger's intercept values were not presented in the figures. I kindly request the authors to include this information in the funnel plot and provide a detailed explanation and interpretation of the results in the results section. Incorporating these findings will strengthen the manuscript's analysis and provide valuable insights into the presence of potential publication bias.

·       I recommend that the authors consider including a more comprehensive interpretation of the results in the discussion section. Although the analysis and evidence were adequately presented, providing a detailed explanation of the results in the discussion would greatly enhance the readers' understanding and interpretation of the systematic review findings. This addition has the potential to offer valuable insights and significantly contribute to the overall clarity and impact of the manuscript. 

Author Response

This review manuscript authored by Hosnieya Labib .et.al aim to provide a comprehensive overview of the prevalence and clinical impact of transition zone anastomosis (TZA) in patients with Hirschsprung disease (HD) who undergo pull-through surgery (PT). The abstract is well-written and effectively conveys the key findings and implications of the study. However, the manuscript requires additional revision to adequately address the issues raised in the review. Further revision of interpretation of results, clarity and accuracy of data analysis and figures could greatly enhance the manuscript impact and make it more engaging for readers.

  1. The valuable systematic review on transition zone anastomosis (TZA) in Hirschsprung disease (HD) provided insightful findings based on studies available until October 2020. However, considering that there might be more recent research articles and case studies on this topic after 2020, I would like to kindly suggest that the authors consider incorporating the latest results into their manuscript. This addition could further enhance the review's comprehensiveness and relevance to the current scientific literature.

First of all, we would like to thank the reviewer for all the feedback and suggestions.
We have considered incorporating the latest results. However, because only 34 studies until 2020 met our inclusion criteria. We don’t expect many more studies that meet our including criteria in the past 2,5 years. And could thereby enhance our review’s comprehensive and relevance. Therefore, we have not updated our search.

  1. Line 75: Please remove “2. Materials and Methods”

We have now removed the double ‘Materials and methods’

  1. In line 108, kindly consider adding clarification within parentheses near the term “rectal biopsy.” Specifically, including “(standard and full-thickness rectal biopsy)” would enhance the clarity of the methods being discussed and provide readers with a better understanding of the types of biopsies considered in the study.

We have now added ‘full thickness biopsy. See lines 111-113: ‘Abnormalities of neuronal innervation had to be confirmed by histopathological examination, either by rectal biopsy (standard or full thickness biopsy) or in the resection specimens.’

  1. In the discussion of histopathological procedures for assessing nerve cell abnormalities, I would like to inquire whether the authors have considered the use of immunohistochemistry with antibodies targeting specific markers like S100, Hu, or calretinin. This approach can be valuable in identifying ganglion cells and aiding in the diagnosis of conditions such as Hirschsprung's disease. Including this possibility in the manuscript could enhance the comprehensiveness of the discussion.

We did not distinguish between the used staining techniques in the different studies. The different staining techniques are outside the scope of this review.

  1. I kindly recommend that the authors consider relocating all supporting/supplementary tables (sTable1 and sTable2) from the main text to a supplementary data  section. The presence of multiple supporting information tables within the main text might potentially confuse readers. By placing these tables in a dedicated section, the manuscript's organization and readability can be improved, allowing readers to access supplementary information more efficiently.

We agree with the reviewer, this presence of multiple supporting information table might confuse the readers. We have now relocated the supporting table after the reference list.

  1. In addition, I would like to suggest including reference numbers in the "authors" column for all the tables. This addition will greatly assist readers in easily identifying the corresponding studies and enhance the overall clarity of the manuscript.

We have now added reference numbers in all the tables.

  1. In page 5, under the section "Quality assessment and risk bias in individual studies," I would like to kindly point out that while the text mentions the Newcastle-Ottawa Scale (NOS) criteria and their respective score ranges (0 to 4, 2, and 3 points), it does not explicitly state the total possible score range for the NOS. Including this information, which ranges from 0 to 9, will provide readers with a clearer understanding of how the total score reflects the methodological quality of the observational studies. In page 5, under the section "Quality assessment and risk bias in individual studies," I would like to kindly bring to the authors' attention that the text mentions the categorization of study quality as "good, fair, or poor" based on the NOS scores. However, specific cutoff values for defining what constitutes "good," "fair," or "poor" quality are not provided. It would be helpful to include this information to clarify the criteria used for categorizing the studies and to enhance the transparency of the quality assessment process.

We have now clarified this. Please see lines 132-135: Good quality: 3 or 4 stars in selection domain AND 1 or 2 stars in comparability domain AND 2. Fair quality: 2 stars in selection domain AND 1 or 2 stars in comparability domain AND 2 or 3. Poor quality: 0 or 1 star in selection domain OR 0 stars in comparability domain OR 0 or 1 stars.’

  1. Furthermore, in the same section, it is mentioned that rating discrepancies were resolved by consensus, which is a standard approach. However, it would be beneficial to provide additional details on how the consensus was reached, such as through discussion, arbitration by a third party, or any other established process. This information would enhance the transparency of the quality assessment process and provide readers with a better understanding of the rigor of the study's methodology.

We agree with the reviewer that we should enhance the transparency for the readers.
Please see lines 135-136: ‘Rating discrepancies were resolved by consensus by including a third party to the discussion.’

  1. I would like to kindly suggest placing the figure number and legends just below each figure in the manuscript. This arrangement will improve the clarity and readability of the figures, making it easier for readers to understand the content and context of each figure.

We have added the numbers above each figure/tables/legends.

  1. Request the authors to include figures with their corresponding legends in the manuscript. Additionally, please ensure that each legend provides a clear and comprehensive explanation of the content depicted in the figure. This addition will enhance the reader's understanding and interpretation of the figures, contributing to the overall clarity and quality of the manuscript.

We agree that each figure/table/legend should include a clear and comprehensive explanation of the content. Each table/figure/legend now includes a number and an interpretation of the results. 

  1. I would like to kindly bring to the authors' attention that in the statistical analysis section, it is mentioned that publication bias was assessed using funnel plot asymmetry and quantified with Egger's intercept. However, the data regarding the Egger's intercept values were not presented in the figures. I kindly request the authors to include this information in the funnel plot and provide a detailed explanation and interpretation of the results in the results section. Incorporating these findings will strengthen the manuscript's analysis and provide valuable insights into the presence of potential publication bias.

At the end of the result section you can find the interpretation of the funnel plot.
Please see lines 243-245: There was no risk of publication bias, based on visual inspection of the funnel plot (see sFigure 1, in the Supporting Information) and based on Egger’s regression (t=0.486, p=0.711).’

  1. I recommend that the authors consider including a more comprehensive interpretation of the results in the discussion section. Although the analysis and evidence were adequately presented, providing a detailed explanation of the results in the discussion would greatly enhance the readers' understanding and interpretation of the systematic review findings. This addition has the potential to offer valuable insights and significantly contribute to the overall clarity and impact of the manuscript. 

We have now made changes to the discussion section. We hope this offers more clarity. 

Reviewer 3 Report

Drs Labib, Roorda and colleagues present a systematic review of outcomes of transition zone anastomosis in Hirschsprung`s disease. 

The review is relevant as it addresses several important clinical outcomes. 

The introduction is clear and gives an overview of the definition and possible clinical implications of TZA. 

The methods section describes the search strategy appropriately. The definitions are clearly explained. The methodology is sound and employs recommended assessment tools, reporting standards and statistical methods recommended for conducting systemic reviews. 

The limitations of the included studies are discussed in detail and point to important aspects for the design of prospective outcome studies. 

The authors discuss the importance of the prospective analysis of cases with standardized definitions and mention the NETS1HD group with core outcome definitions. The same group have published the results of a prospective cohort study shortly after the last search (October 2020 , November 2020; Allin BSR et al. Archives Dis Childhood 2020 Nov 2; 106(5):484-490. ). This could be included in the discussion. 

formal aspects:

L75: formatting 

L81: ..histopathologically confirmed... 

L230: ..as the studies only described...

L296: ...diarrhea.. 

L297: better: damage to sphincter complex

L309: restructure sentence e.g. : To improve outcomes and aid in predicting the need for redo pull-through procedures it is important to detect TZA early. 

L327: suggest: toilet training 

L334: ..peri-operative [typo]

L335: suggest: Due to the poor quality of the studies the findings are difficult to generalize. 

or: The quality of the included studies reduces the generalizability of the findings. 

Minor adjustments and suggestions could be included (please see above). 

Author Response

Drs Labib, Roorda and colleagues present a systematic review of outcomes of transition zone anastomosis in Hirschsprung`s disease. 

The review is relevant as it addresses several important clinical outcomes. 
The introduction is clear and gives an overview of the definition and possible clinical implications of TZA. 
The methods section describes the search strategy appropriately. The definitions are clearly explained. The methodology is sound and employs recommended assessment tools, reporting standards and statistical methods recommended for conducting systemic reviews. 
The limitations of the included studies are discussed in detail and point to important aspects for the design of prospective outcome studies. 
The authors discuss the importance of the prospective analysis of cases with standardized definitions and mention the NETS1HD group with core outcome definitions. The same group have published the results of a prospective cohort study shortly after the last search (October 2020 , November 2020; Allin BSR et al. Archives Dis Childhood 2020 Nov 2; 106(5):484-490. ). This could be included in the discussion. 

formal aspects:

L75: formatting 

L81: ..histopathologically confirmed... 

L230: ..as the studies only described...

L296: ...diarrhea.. 

L297: better: damage to sphincter complex

L309: restructure sentence e.g. : To improve outcomes and aid in predicting the need for redo pull-through procedures it is important to detect TZA early. 

L327: suggest: toilet training 

L334: ..peri-operative [typo]

L335: suggest: Due to the poor quality of the studies the findings are difficult to generalize. 

or: The quality of the included studies reduces the generalizability of the findings. 

Minor adjustments and suggestions could be included (please see above). 

Thank you for your feedback and suggestions. We have corrected the adjustments mentioned above.

Reviewer 4 Report

The authors provide a systematic review of the literature about transition zone anastomosis in patients with Hirschsprung´s disease. Their systematic approach is concise and the conclusions a clear. However, I have some concerns regarding the language, grammar and format of the provided manuscript. Here are my comments:

Language

The authors should check the manuscript for language and format issues. Some examples are listed below. I recommend to get help by a native speaker. If none is available use software like “Grammarly” to screen the whole manuscript. The basic screen is free and will improve the quality of the manuscript….

1. “Different operation techniques can be used to restore bowel continuity, either by creating a pouch, which is used in Duhamel. Or by creating a straight anastomosis, used in all other techniques.” Page 2 line 52

-       One or two sentences?

2. “Incomplete removal of the transition zone and/or aganglionic segment, called a transition zone anastomosis (TZA).” Page 2 line 54

-       This is not a complete sentence

3. “Localization” page 2 line 57

-       Localization or localisation (both are possible; first is American second British, please choose either one for the whole manuscript and change it accordingly).

4. “Analyzing” page 2 line 61

-       Again, see above, its best to choose one and stick to it

5. “Material and Methods” page 2 line 75#

-       This is displaced and doubled

6. “diarheaw” page 14, line 296

These are just some examples, mostly in the introduction. I recommend to use online software (see above) to check the whole manuscript (it doesn´t need much time!).

Content

1. „The aggregated event rate of the occurrence of a TZA was 9% (ER=0.09, 95%CI=0.05-0.14, p<0.001, I2=86%). The aggregated event rate of the occurrence of a TZA was 25% (ER=0.25, 95%CI=0.16 -0.37, 170 p<0.001, I2=92%).“

-       The second statement is probably meant to be placed on page 8 in line 166?

2. Overall the authors should precisely state in the figure legend what the forest plot is showing. Is it comparing the group redo with initial pull through? Is the forest plot really the graphical illustration of choice here? Mostly forest plots are used to show which procedure or which outcome variable is superior to another. But comparing a retrospective prevalence of a finding? I think the authors should consider to choose somethingelse to make clear what they want to show…

3. The authors state that there are less TZA cases in patients after Duhamel. As Duhamel procedure leaves a relevant part of aganglionic bowel for the pouch does it even make sense to use the term TZA in patients after Duhamel procedure? I would like to hear the authors opinion on that.

The authors present a study on a highly relevant topic. But for now, the manuscript cannot be accepted until the mentioned points are addressed.

The authors provide a systematic review of the literature about transition zone anastomosis in patients with Hirschsprung´s disease. Their systematic approach is concise and the conclusions a clear. However, I have some concerns regarding the language, grammar and format of the provided manuscript. Here are my comments:

Language

The authors should check the manuscript for language and format issues. Some examples are listed below. I recommend to get help by a native speaker. If none is available use software like “Grammarly” to screen the whole manuscript. The basic screen is free and will improve the quality of the manuscript….

1. “Different operation techniques can be used to restore bowel continuity, either by creating a pouch, which is used in Duhamel. Or by creating a straight anastomosis, used in all other techniques.” Page 2 line 52

-       One or two sentences?

2. “Incomplete removal of the transition zone and/or aganglionic segment, called a transition zone anastomosis (TZA).” Page 2 line 54

-       This is not a complete sentence

3. “Localization” page 2 line 57

-       Localization or localisation (both are possible; first is American second British, please choose either one for the whole manuscript and change it accordingly).

4. “Analyzing” page 2 line 61

-       Again, see above, its best to choose one and stick to it

5. “Material and Methods” page 2 line 75#

-       This is displaced and doubled

6. “diarheaw” page 14, line 296

These are just some examples, mostly in the introduction. I recommend to use online software (see above) to check the whole manuscript (it doesn´t need much time!).

Content

1. „The aggregated event rate of the occurrence of a TZA was 9% (ER=0.09, 95%CI=0.05-0.14, p<0.001, I2=86%). The aggregated event rate of the occurrence of a TZA was 25% (ER=0.25, 95%CI=0.16 -0.37, 170 p<0.001, I2=92%).“

-       The second statement is probably meant to be placed on page 8 in line 166?

2. Overall the authors should precisely state in the figure legend what the forest plot is showing. Is it comparing the group redo with initial pull through? Is the forest plot really the graphical illustration of choice here? Mostly forest plots are used to show which procedure or which outcome variable is superior to another. But comparing a retrospective prevalence of a finding? I think the authors should consider to choose somethingelse to make clear what they want to show…

3. The authors state that there are less TZA cases in patients after Duhamel. As Duhamel procedure leaves a relevant part of aganglionic bowel for the pouch does it even make sense to use the term TZA in patients after Duhamel procedure? I would like to hear the authors opinion on that.

The authors present a study on a highly relevant topic. But for now, the manuscript cannot be accepted until the mentioned points are addressed.

Author Response

The authors provide a systematic review of the literature about transition zone anastomosis in patients with Hirschsprung´s disease. Their systematic approach is concise and the conclusions a clear. However, I have some concerns regarding the language, grammar and format of the provided manuscript. Here are my comments:

 Language

The authors should check the manuscript for language and format issues. Some examples are listed below. I recommend to get help by a native speaker. If none is available use software like “Grammarly” to screen the whole manuscript. The basic screen is free and will improve the quality of the manuscript….

  1. “Different operation techniques can be used to restore bowel continuity, either by creating a pouch, which is used in Duhamel. Or by creating a straight anastomosis, used in all other techniques.” Page 2 line 52

-       One or two sentences?

  1. “Incomplete removal of the transition zone and/or aganglionic segment, called a transition zone anastomosis (TZA).” Page 2 line 54

-       This is not a complete sentence

  1. “Localization” page 2 line 57

-       Localization or localisation (both are possible; first is American second British, please choose either one for the whole manuscript and change it accordingly).

  1. “Analyzing” page 2 line 61

-       Again, see above, its best to choose one and stick to it

  1. “Material and Methods” page 2 line 75#

-       This is displaced and doubled

  1. “diarheaw” page 14, line 296

These are just some examples, mostly in the introduction. I recommend to use online software (see above) to check the whole manuscript (it doesn´t need much time!).

We would like to thank the reviewer for the feedback and suggestions.
We have now changed all the language examples listed above. And we have also used the software ‘Grammarly’ to screen  the manuscript.

Content

  1. „The aggregated event rate of the occurrence of a TZA was 9% (ER=0.09, 95%CI=0.05-0.14, p<0.001, I2=86%). The aggregated event rate of the occurrence of a TZA was 25% (ER=0.25, 95%CI=0.16 -0.37, 170 p<0.001, I2=92%).“

-       The second statement is probably meant to be placed on page 8 in line 166?

We have corrected this. Please see lines 177-182: ‘After exclusion of the studies that only included patients who underwent redo pull-through, 19 studies were included in the meta-analysis on the prevalence of TZA, representing 1817 patients. The aggregated event rate of the occurrence of a TZA was 9% (ER=0.09, 95%CI=0.05-0.14, p<0.001, I2=86%). When including all studies on the prevalence of TZA, the aggregated event rate of the occurrence of a TZA was 25% (ER=0.25, 95%CI=0.16 -0.37, p<0.001, I2=92%).’

  1. Overall the authors should precisely state in the figure legend what the forest plot is showing. Is it comparing the group redo with initial pull through? Is the forest plot really the graphical illustration of choice here? Mostly forest plots are used to show which procedure or which outcome variable is superior to another. But comparing a retrospective prevalence of a finding? I think the authors should consider to choose something else to make clear what they want to show…

Our forest plot indeed compares the prevalence of TZA in studies only including patients after initial pull-through to studies that include patients who underwent redo pull-through. The reason we chose a forest plot is because a forest plot gives a graphic representation of the findings of multiple studies investigating the same outcome. The purpose of our forest plot is to show the readers that there is a significant difference when including only studies that include redo cohorts compared to studies including patients after initial pull-through.

Please see lines 185-188: ‘There was a significant difference in the aggregated event rate of TZA between studies with only patients who underwent redo pull-through, compared to studies with patients after initial pull-through (Q=49.9, p<0.001).’

  1. The authors state that there are less TZA cases in patients after Duhamel. As Duhamel procedure leaves a relevant part of aganglionic bowel for the pouch does it even make sense to use the term TZA in patients after Duhamel procedure? I would like to hear the authors opinion on that.

Thank you for your valuable comment. We agree with you that it does not sound very logical. With the Duhamel technique a pouch is  created  with an end-to-side anastomosis with  aganglionic rectal tissue on the ventral side and ganglionictissue on the dorsal side. The dorsal, presumed ganglionic, side is used in the pull-through, therefore it is relevant to prevent TZA in the Duhamel pouch. TZA on the dorsal (proximal) side of the anastomosis, can also result in obstructive symptoms. If aganglionic or hypoganlionic tissue is anastomosed with the pouch there is a TZA.

The authors present a study on a highly relevant topic. But for now, the manuscript cannot be accepted until the mentioned points are addressed.

Round 2

Reviewer 1 Report

In the revised manuscript, the authors have addressed all my concerns and comments. The study has merit and will be of interest to our readers. I would like to congratulate the authors for their work.

Minor editing needed

Author Response

Thank you so much. Our manuscript is improved because of your feedback and suggestions. 

Kind regards,

Hosnieya Labib and all authors 

Reviewer 2 Report

This revised manuscript authored by Hosnieya Labilb et.al aim to provide a comprehensive overview of the prevalence and clinical impact of transition zone anastomosis (TZA) in patients with Hirschsprung disease (HD) who undergo pull-through surgery (PT). After conducting a thorough review of the revised manuscript, I have concerns regarding the provided figure.

Please incorporate the Egger's intercept values into the figure (sFigure1) as a legend. While I acknowledge that the Egger's intercept value has been discussed in the text, I had previously suggested to the authors to include this value as a legend in the plot. This inclusion would enhance the clarity of the figures for readers, eliminating the need to repeatedly refer back and forth between the text and the visual representation in the manuscript.

Author Response

Thank you for your suggestion. We have now added Egger’s intercept value (mentioned in the text) in the sFigure1, so the reader does not have to refer back and forth between the text and figure.

Reviewer 4 Report

The authors addressed or explained all comments.

I think the article should be accepted if the recommendations of the other reviewers were considered.

Author Response

(The authors gave the same response as above.)
